# Two-Step Upcycling Process of Lignocellulose into Edible Bacterial Nanocellulose with Black Raspberry Extract as an Active Ingredient

**DOI:** 10.3390/foods12162995

**Published:** 2023-08-09

**Authors:** Marijana Ponjavic, Vuk Filipovic, Evangelos Topakas, Anthi Karnaouri, Jelena Zivkovic, Nemanja Krgovic, Jelena Mudric, Katarina Savikin, Jasmina Nikodinovic-Runic

**Affiliations:** 1Institute of Molecular Genetics and Genetic Engineering, University of Belgrade, Vojvode Stepe 444a, 11000 Belgrade, Serbia; mponjavic@imgge.bg.ac.rs (M.P.); vfilipovic@imgge.bg.ac.rs (V.F.); 2Industrial Biotechnology and Biocatalysis Group, Biotechnology Laboratory, School of Chemical Engineering, National Technical University of Athens, Zografou Campus, 5 Iroon Polytechniou Str., 15772 Athens, Greece; vtopakas@chemeng.ntua.gr; 3Laboratory of General and Agricultural Microbiology, Department of Crop Science, Agricultural University of Athens, 11855 Athens, Greece; akarnaouri@aua.gr; 4Institute for Medicinal Plants Research “Dr Josif Pančić”, Tadeuša Košćuška 1, 11000 Belgrade, Serbia; jzivkovic@mocbilja.rs (J.Z.); nkrgovic@mocbilja.rs (N.K.); jmudric@mocbilja.rs (J.M.); ksavikin@mocbilja.rs (K.S.)

**Keywords:** lignocellulose, beechwood biomass, bacterial nanocellulose, cyanidin-3-O-rutinoside, release kinetics, upcycling

## Abstract

(1) Background: Bacterial nanocellulose (BNC) has gained in popularity over the years due to its outstanding properties such as renewability, biocompatibility, and bioavailability, and its use as an eco-friendly material of the future for replacing petrochemical products. (2) Methods: This research refers to the utilization of lignocellulose coming from wood waste via enzymatic hydrolysis to produce biopolymer BNC with an accumulation rate of 0.09 mg/mL/day. Besides its significant contribution to the sustainability, circularity, and valorization of biomass products, the obtained BNC was functionalized through the adsorption of black raspberry extract (BR) by simple soaking. (3) Results: BR contained 77.25 ± 0.23 mg GAE/g of total phenolics and 27.42 ± 0.32 mg CGE/g of total anthocyanins. The antioxidant and antimicrobial activity of BR was evaluated by DPPH (60.51 ± 0.18 µg/mL) and FRAP (1.66 ± 0.03 mmol Fe^2+^/g) and using a standard disc diffusion assay, respectively. The successful synthesis and interactions between BNC and BR were confirmed by FTIR analysis, while the morphology of the new nutrient-enriched material was investigated by SEM analysis. Moreover, the in vitro release kinetics of a main active compound (cyanidin-3-O-rutinoside) was tested in different release media. (4) Conclusions: The upcycling process of lignocellulose into enriched BNC has been demonstrated. All findings emphasize the potential of BNC–BR as a sustainable food industry material.

## 1. Introduction

Cellulose, a long-chain polysaccharide, is commonly distributed in nature as the essential and most abundant natural polymer on the planet and a key source of renewable materials on an industrial scale [1]. Cellulose can be obtained from various sources such as the cell wall of plants and woods, seaweed, algae, and tunicates [2]. The annual production of cellulose is about 100 billion tons [3]. Cellulose and its derivatives are extensively utilized in various fields, primarily due to their renewable nature and biocompatibility. Among these applications, their use as biomaterials stands out as particularly notable. Cellulose can also be produced by fungi and some species of non-pathogenic bacteria, with one the most efficient and studied being an aerobic Gram-negative bacteria *Komagateibacter xylinus* [4]. Several strains of *K. xylinus* produce extracellular cellulose as a protective biofilm against undesirable external influences [5].

Bacterial cellulose is produced using a simple production process and it is usually in the form of fibrils of nano-dimensions (BNC). Its principal benefits are nontoxicity, high purity, and biocompatibility, along with high crystallinity, a large surface area (leading to high porosity and water-holding capacity), and enhanced tensile strength [6,7]. The mechanical properties, crystallinity, and shape of the BNC obtained depend on oxygen delivery, pH variations, and other fermentation process conditions [6]. It should be also mentioned that BNC can be manufactured through the bottom-up technique from lignocellulosic biomass [8]. Due to its appropriate rheological properties and biodegradability, as well as its potential to form hydrogels, BNC is used in medicine [9] and cosmetics [10,11]. and in the food industry as a sweet bacterial cellulose gel, called nata de coco. For this delicacy from the Philippines, coconut water is used as a substrate for BNC production [12]. Furthermore, nanocellulose can interact with fatty foods to inter-mediate with the activity of pancreatic lipase, substantially reducing fat digestion and absorption in the organism [13]. On the other side, nanocellulose has no significant impact on body weight gain, food intake, and blood glucose homeostasis, and its consumption has no influence on small intestinal morphology [14].

Black raspberry (*Rubus occidentalis* L.), a member of the family *Rosaceae*, is a deciduous shrub native to eastern North America [15]. Due to its dietary importance and significant economic value, it is now cultivated in other parts of the world [16,17]. In recent years, many studies have shown that black raspberry possesses various pharmacological effects [18,19,20] that are attributed to a range of secondary metabolites—flavonoids (anthocyanins, flavanols and flavonols), phenolic acids (ellagic and protocatechuic), and ellagitannins [21,22]. However, anthocyanins, particularly cyanidin derivatives, are identified as the major compounds offering these benefits, as they are accumulated in high amounts in black raspberry [23,24]. Not only fruits but also black raspberry fruit pomaces represent a promising plant material with nutritional and health-promoting properties. It is well-documented that a large volume of waste, berry fruit pomace, is produced during the processing of fresh berries at industrial levels, causing significant ecological and financial problems [25,26]. Bearing this in mind, great scientific interest focuses on its reutilization as a source of polyphenols along with other beneficial components (vitamins, minerals, and fatty acids) [26].

One of main limitations of microbial fermentation processes lies in the high costs of nutrients, including, among others, carbon and nitrogen sources that are required for cell growth and the synthesis of the desired products [27]. Therefore, sugars obtained from lignocellulosic residues that are by-products of the forest industry and often remain underutilized or are burned for energy recovery with a very low sale price offer many advantages. Apart from rendering the overall process more profitable, this approach allows for the sustainable use and valorization of biomass resources for the production of high-value bulk products, such as fuels, chemicals, additives, polymers, and others, thus promoting a circular economy [24,28]. Lignocellulosic-derived sugars can be employed as a carbon source in order to produce bacterial nanocellulose of high quality through fermentation [8]. Within the frame of a biorefinery concept, the possibility of taking advantage of all sugar streams obtained from lignocellulose treatment for the synthesis of bacterial nanocellulose can lead to the maximization of product yield and elimination of sugar losses, also being in concert with a holistic and zero-waste use of sugar-streams from natural resources. The utilization of lignocellulose-derived sugar streams requires a primary pretreatment step which not only renders the biomass more amenable to subsequent enzymatic saccharification and microbial fermentation, but also achieves the fractionation of the biomass to its three main structural components, namely, cellulose, hemicellulose, and lignin [8,29]. In this study, a sugar-rich hydrolysate from organosolv-pretreated beechwood was used as a carbon source for the fermentation of *Komagataeibacter medellinensis* towards the production of nanocellulose.

The main scope of this study was to replace commonly used glucose with the lignocellulose-derived sugars obtained from beechwood biomass for bacterial nanocellulose growth. Following this approach, waste products of the wood industry are upcycled into a highly valuable biopolymer, namely, bacterial nanocellulose. One step further, the pulpy residue remaining after fruit extraction and juice preparation, black raspberry in this case, can be used to enrich the obtained bacterial nanocellulose through simple adsorption. Therefore, a new enriched bioproduct with antimicrobial and antioxidant properties is generated.

## 2. Materials and Methods

### 2.1. Materials and Chemicals

Potassium hydroxide (KOH) was supplied from Fisher Chemicals (Fisher Scientific, Loughborough, Leicestershire, UK), while yeast extract and bactopeptone were ordered from Biolife (Milano, Italy) and sodium hydrogen phosphate heptahydrate (Na_2_HPO_4_·7 H_2_O) was product of Acros Organics (Verona, Spain). Acetonitrile was purchased from Merck (Darmstadt, Germany), formic acid from Sigma Aldrich (Munich, Germany), cyanidin-3-O-rutinoside from Extrasynthese (Genay, France), ethanol and sodium carbonate anhydrous (Na_2_CO_3_) from Centrohem (Stara Pazova, Serbia), and hydrochloric acid (HCl; 36.2%) from Zorka Pharma (Sabac, Serbia). Folin–Ciocalteu reagent was purchased from Carlo Erba Reagents (Emmendingen, Germany), sodium carbonate anhydrous (Na_2_CO_3_) from Centrohem (Stara Pazova, Serbia), and gallic acid, 2,2-dyphenyl-1-picrylhydrazyl (DPPH), ascorbic acid, sodium acetate, 2,4,6-tris(2-pyridyl)-(S)-triazine (TPTZ), sodium acetate trihydrate, iron(III) chloride hexahydrate (FeCl_3_∙6H_2_O), iron(II) sulfate heptahydrate (FeSO_4_∙7H_2_O), and citric acid were purchased from Sigma Aldrich (Munich, Germany).

### 2.2. Black Raspberry Extraction

#### 2.2.1. Black Raspberry Pomace

Fully ripe fruits of *R. occidentalis* were collected from a private plantation at Povlen (44°07′50″ N, 19°44′25″ E), mountain in western Serbia, in July of 2020. The black raspberry pomace was prepared by pressing the fresh berries in a commercial juicer (Philips Aluminium Collection Juicer HR1865/00, Philips, Amsterdam, The Netherlands), and then dried in a laboratory drier (FDK24DW, Gorenje, Ljubljana, Slovenia) at 40 °C for 3 days.

#### 2.2.2. Black Raspberry Pomace Extraction

Black raspberry pomace was ground in a laboratory mill (DēLonghi, Treviso, Italy) and sieved through a set of sieves to obtain 0.75–2 mm particle fractions, following the procedure of the 5th Yugoslavian Pharmacopoeia [30]. Pulverized black raspberry pomace was extracted with 60% ethanol aqueous solution (1:2, *w*/*v*) using percolation, which lasted for 24 h at room temperature. Afterward, extract was frozen at −50 °C (Arctiko ULUF 65 Benchtop/Undercounter Freezer, Esbjerg, Denmark) and lyophilized (Beta 1–8 Freeze Dryer, Martin Christ, GmbH, Osteroide am Harz, Germany). Lyophilized black raspberry pomace extract (BR) was stored in a dry room until analysis.

### 2.3. Preparation of a Sugar-Rich Hydrolysate from Pretreated Beechwood Biomass

Beechwood (Lignocel**^®^** HBS 150–500, JRS GmbH and Co KG, Rosenberg, Germany) was subjected to acid-free mild oxidative organosolv fractionation (OxiOrganosolv) with a solution of 50:50 (*w*/*w*) H_2_O:acetone in the liquid phase, at a liquid-to-solid ratio of 10, at 160 °C for 120 min with an O_2_ pressure of 16 bar, as previously described [31]. After pretreatment, the solid pulp was obtained through vacuum filtration, washed, and air-dried. The composition of the recovered solid fraction was 76.6 wt% cellulose, 13.3 wt% hemicellulose, and 3.2 wt% lignin [31]. Enzymatic hydrolysis took place for 72 h, in 1 L flasks, at 50 °C, under 160 rpm agitation using Cellic**^®^** CTec2 from Novozymes A/S (Bagsværd, Copenhagen, Denmark), in 25 mM phosphate-citrate buffer pH 5.0. The initial concentration of solids was 9 wt% and the enzyme loading was 9 mg/g substrate with total reaction volume of 500 mL. At the end of hydrolysis, solids were removed via centrifugation and hydrolysate was collected, concentrated 2 times with freeze drying, and stored until further use. The total sugar content of the hydrolysate that was further used as a carbon source in microbial fermentations was 61.5 mg/mL, as determined by the 3,5-dinitrosalicylic acid (DNS) assay [32], while the concentration of glucose was 49.5 mg/mL, as calculated with the glucose oxidase/peroxidase method (GOD/POD) [33].

### 2.4. Bacterial Nanocellulose Production

Bacterial nanocellulose (BNC) was produced by cultivation of *K. medellinensis* ID13488 strain (CECT 8140 (Spanish Type Culture collection) in a modified Hestrin–Schramm (HS) medium consisting of 20 g/L of sugar-rich hydrolysate (containing 61.5 and 49.5 mg of total sugars and glucose/mL of hydrolysate, respectively, as was described in Section 2.2), 5 g/L peptone, 5 g/L yeast extract, 2.5 g/L Na_2_HPO_4_, and 1.15 g/L citric acid, at pH 4.5 under static conditions. Cultures were incubated for 7 and 30 days to obtain different thickness of the material. BNC was collected, treated with potassium hydroxide (KOH, 5 wt%) for 12 h, and washed with deionized H_2_O to achieve the neutral pH. The produced BNC in the form of gel was further used for preparation of active compound containing formulation with black raspberry extract, BR (BNC-BR). For comparison purposes, control BNC was also lyophilized till constant mass. White, fine material was obtained.

The BNC yield was calculated as followed:(1)$$Yield\left(mg/mL\right)=\frac{m_{dryBNC}}{V_{culturemedium}}$$
where the m_dry_ is dry weight of purified BNC membranes per 1 mL of culture medium.

The productivity was calculated according to Equation (2):(2)$$Productivity=\frac{Yield}{Culture\;time}$$
where productivity was determined as the obtained BNC yield divided by the culture time (mg/mL/day).

### 2.5. Bacterial Nanocellulose–Black Raspberry (BNC-BR) Film Preparation

Bacterial nanocellulose containing black raspberry as an active compound was prepared as follows: 40.0 g of wet BNC (in the form of gel, containing 90% of water, which means approximately 4.00 g of dry BNC) was soaked in 30 mL of distilled water containing 400 mg of black raspberry extract (10 wt% calculated to the amount of dry BNC) and left at stirring conditions at room temperature for 24 h. The prepared active material was casted on a Petri dish (diameter of 20 cm) and dried in an oven at 37 °C till most of water was evaporated when the BNC-BR film was lyophilized till constant mass was reached. A reddish, fine, compact film was obtained and further characterized, as described below.

#### Efficiency of Black Raspberry Extract Adsorption on BNC

The efficiency of adsorption of black raspberry extract as an active compound was investigated both by gravimetrical analysis and UV-VIS spectroscopy (*λ* = 520 nm). Considering the physical masses of BNC and black raspberry extract in feed and after BNC-BR preparation, it was possible to estimate the efficiency of adsorption (EA%) by applying Equation (3):(3)$$EA(\%)=\frac{m_{BR}}{m_{BR+BNC}}\times 100$$
where m_BR_ refers to the mass of black raspberry extract added in feed, while m_BR+BNC_ defines the final mass of the obtained material (BNC-BR).

Additionally, the commonly used procedure of defining EA% by UV-VIS spectroscopy when an active compound is adsorbed in a material was used, where the concentration of black raspberry extract was measured before and after adsorption. For this purpose, a calibration curve of black raspberry extract in water was prepared. For this purpose, the concentrations ranging from 1000 ppm to 3000 ppm were measured at wavelength of *λ* = 520 nm.

### 2.6. Characterization of Obtained BNC-BR Film

#### 2.6.1. Fourier-Transform Infrared Spectroscopy Analysis (ATR-FTIR)

The Fourier-transform infrared spectroscopy (FTIR) was used not only to identify the structure of BNC and BR extract, but also to detect the possible interaction between BNC and BR extract after their mixing. The samples were measured using IR-Affinity spectrophotometer (Thermo Fisher Scientific, NICOLET iS10, Waltham, MA, USA) in an attenuated total reflection (ATR) mode. All the measurements were performed in the predefined wavenumber range of 4000 to 400 cm^−1^, at room temperature and a resolution of 4 cm^−1^, with the fixed number of scans at 32.

#### 2.6.2. Optical Microscopy (OM) and Scanning Electron Microscopy Analysis (SEM)

Prior to the samples being submitted to SEM analysis, bacterial nanocellulose and bacterial nanocellulose with the adsorbed black raspberry extract were recorded by optical microscopy using Olympus SZX10 attached to a digital imaging system. The samples were captured at different magnifications.

In addition, SEM analysis was employed not only to examine the surface morphology of bacterial nanocellulose before and after the addition of black raspberry extract, but also to detect the possible morphological changes caused by presence of active compound. SEM micrographs were captured using JEOL JSM-6390LV SEM (JEOL USA Inc., Peabody, MA, USA), and recorded at voltage of 10–15 kV. Dry, lyophilized samples were supported on carbon tape, coated with a conducting layer of gold and submitted to analysis.

### 2.7. In Vitro Release Study

The release profile of the marker compound (cyanidin-3-O-rutinoside—CR) from lyophilized BR was compared with the release profile of the BNC-BR sample (in 0.5, 5, 10, 15, 30, 60, 120, and 180 min) under the conditions reported in 10th European Pharmacopoeia [34], with slight modifications. The prepared BNC-BR sample was tested in four media to assess the influence of pH value (1.2, 4.5, 6.8, and 7.4) on the CR release profile (in 0.5, 5, 10, 15, 30, and 60 min). In addition, the influence of the samples’ thickness was also investigated by comparing the release profile of the BNC-BR sample with a thickness of 150 μm (BNC-BR 150), with the release profile of the BNC-BR sample with a thickness of 250 μm (BNC-BR 250), at pH 4.5. The following time points were used: 0.5, 5, 10, 15, 30, and 60 min. In vitro release study was conducted in the beakers placed in the water bath (Erweka DT70 (Erweka, Langen, Germany), at 100 rpm, at 37 ± 1 °C. The small volumes of medium were withdrawn and filtrated at predefined time intervals and immediately replaced with a fresh portion. The concentration of dissolved marker compounds was determined by high-performance liquid chromatography (Section 2.9). All the measurements were performed in triplicate and the results are presented as the mean value ± standard deviation (S.D.). The comparison of CR release profiles was conducted by calculating the similarity factor (*f*_2_), according to Equation (4). A value *f*_2_ lower than 50 implies that the profiles are significantly different, while the *f*_2_ value of 50 corresponds to an average difference of 10% at all specified time points:(4)$$f_{2}=50\times log\left\{{\left[1+\frac{1}{n}\sum_{n=1}^{t}{\left(R_{t}-T_{t}\right)}^{2}\right]}^{-0.5}\times 100\right\}$$
where *n* is the number of dissolution sampling times, and *R_t_* and *T_t_* are the active compound (gentiopicroside) release percentages at each time for the reference and test sample, respectively.

### 2.8. Kinetic Modeling of Cyanidin-3-O-Rutinoside Release

Release kinetics of CR at different pH conditions (1.2, 4.5, 6.8, and 7.4) were evaluated by using 60 min dissolution profile. The data were fitted into zero-order, first-order, Higuchi, and Korsmeyer–Peppas models (Table 1). The model with the highest correlation coefficient (*R*^2^) was selected as the model which describes release kinetics through the most realistic mechanism.

### 2.9. High-Performance Liquid Chromatography Analysis (HPLC)

In order to determine individual anthocyanin content, namely, CR, samples were analysed using an Agilent 1260 HPLC system (Agilent Technologies, Santa Clara, CA, USA) equipped with Lichrospher RP-18 column (250 × 4.0 mm; 5.0 µm particle size) and DAD detector. The mobile phase contained 1% orthophosphoric acid in water (phase A) and pure acetonitrile (phase B). Gradient elution program was as follows: 0–8% B (6 min), 8–10% B (9 min), and 10–25% B (5 min). Total run time was 20 min, injection volume 10 µL, flow rate 0.8 mL/min, and column temperature 25 °C. The wavelength on which chromatograms were recorded was 520 nm. Identification was performed by comparing the retention time and UV spectrum of targeted compound with those obtained from the reference standard, while, for the quantification, external calibration curve was used (concentration range: 0.05–0.40 mg/mL; y=6088.6x−11.371, *R*^2^ = 0.9998).

### 2.10. Total Phenolic Content (TPC)

The TPC in BR was determined using the Folin–Ciocalteu (FC) spectrophotometric method [38]. Briefly, extract dissolved in 60% ethanol (100 µL) was mixed with ten-fold-diluted FC reagent (750 µL). After 5 min, 60 g/L Na_2_CO_3_ solution (750 µL) was added. The absorbance was measured at 725 nm after 90 min incubation in the dark, at room temperature. The result was expressed as milligrams of gallic acid equivalents per gram of dry weight of extract (mg GAE/g).

### 2.11. Total Anthocyanin Content (TAC)

The TAC in BR was estimated following the procedure described in 10th European Pharmacopoeia with slight modifications [34]. Dried extract (2.5 mg) was dissolved in 5 mL of 60% ethanol. A 10-fold dilution of this solution was prepared in 0.1% hydrochloric acid in methanol. The absorbance was recorded at 528 nm using 0.1% hydrochloric acid in methanol as a compensating solution. The result was expressed as mg of cyanidin-3-O-glucoside equivalent per g of dry weight of extract (mg CGE/g).

The percentage content of anthocyanins was calculated from the expression:(5)Content%=A×5000718×m×100
where *A* is the absorbance at 528 nm; 718 represents specific absorbance of cyanidin-3-glucoside chloride at 528 nm; and *m* is the mass of the substance to be examined in grams.

### 2.12. DPPH Radical Scavenging Activity

The adapted 2,2-diphenyl-1-picrylhydrazyl (DPPH) assay described by Kolundžić [39] was used for evaluation of radical scavenging activity of BR. A dilution series of methanolic solution of BR (100 µL) were mixed with 0.5 mM solution of DPPH in methanol (500 µL), vigorously shaken, and incubated at room temperature in a dark place for 30 min. The absorbance was measured at 517 nm, against methanol as a blank. The control represented the mixture of methanol and DPPH solution. The percentage inhibition of DPPH radicals was calculated using Formula (6):(6)$$DPP\;Hinhibition\left(\%\right)=\left[\left(AC-AS\right)/AC\right]\times 100$$
where *AS* was absorbance of sample solution; and *AC* refers to the absorbance of control solution.

Ascorbic acid was used as positive control. The results were expressed as *IC*_50_ values.

### 2.13. Ferric-Reducing Antioxidant Power (FRAP) Assay

The reducing capacity (ability to reduce Fe^3+^ to Fe^2+^) of BR was investigated following the procedure for FRAP assay of *Benzie and Strain* [40]. Prior to the analysis, FRAP reagent, which contained 300 mmol/L of sodium acetate buffer (pH 3.60), 10 mmol/L of 2,4,6-tris(2-pyridyl)-(*S*)-triazine (TPTZ) in 40 mmol/L HCl, and 20 mmol/L FeCl_3_∙6H_2_O solution (10:1:1, *v*/*v*/*v*), was prepared. Then, 100 µL of the previously diluted BR was added to 3 mL of FRAP reagent. After 30 min incubation at 37 °C, the absorbance was measured at 593 nm. Methanol was used instead of extract to prepare blank, while the ascorbic acid was used as positive control. The results were expressed as mmol Fe^2+^/g of dry weight of extract, using the calibration curve (y=0.0209x+0.0015; *R*^2^ = 0.9976).

The spectrophotometric readings were conducted on Cary 3500 Multicell UV-Vis Spectrophotometer (Agilent Technologies, Santa Clara, CA, USA). All measurements were performed in triplicates, and the results were expressed as mean value ± standard deviation.

### 2.14. Antimicrobial Assay

The antimicrobial activity of the BR extract were determined against *Candida albicans* ATCC 10231 and a panel of bacteria (all obtained from the National Collection of Type Cultures (NCTC) and the American Type Culture Collection (ATCC)) including *Staphylococcus aureus* NCTC 6571, *Pseudomonas aeruginosa* ATCC 10332, *Listeria monocytogenes NCTC 11994*, and *Escherichia coli* NCTC 2001 using standard disc diffusion assay with 1000, 500, 250, and 100 μg of compound per disc. Plates were incubated at 37 °C for 24 h after which the growth inhibition zones were measured. Blank disc impregnated with sterile water was used as control (10 µL/disc).

### 2.15. Statistical Analysis

The experimental data are presented as the mean values ± standard deviation at three repeats. Obtained results were statistically analyzed by Statistica software version 5.0 (StatSoft Co., Tulsa, OK, USA). Significances of differences among samples were analyzed by Tukey’s test. Differences at *p* < 0.05 are considered significant.

## 3. Results and Discussion

Bacterial nanocellulose (BNC) is a linear homopolymer consisting of *β*-D-glucopyranose monomers connected with *β*-1,4-glycosidic linkages (the repeating unit is the disaccharide cellobiose) [41]. BNC containing black raspberry extract (BNC-BR) as an active compound was successfully produced by utilizing a glucose-rich enzymatic hydrolysate from pretreated beechwood as a carbon source. The overall concept of upcycling lignocellulosic residues into highly valuable bacterial nanocellulose is presented in Figure 1. Commonly used glucose for BNC production was substituted by forest-biomass-derived sugars, leading to a high BNC yield with a value of 1.3 mg per 1 mL of culture medium and a productivity that reached 0.09 mg/mL/day. When pure glucose was used as a carbon source, a higher yield of BNC was achieved (6 g/L) [42], while BNC produced from agro-industrial wastes were comparable at 1.5–2.6 g/L [43]. Recently, *Bacillus* spp. that were very efficient in cellulolytic activity were reported, that could potentially make this process even greener, as they could be used as biocatalysts to produce sugar-rich lignocellulose hydrolysates [44,45].

The BNC produced in this way was utilized for the adsorption of black raspberry extract by a simple soaking method, resulting in the BNC-BR material (yielded approx. 70%). The efficiency of adsorption was 56.3 ± 1.9% when defined by gravimetric analysis, and EA% of 47.0 ± 1.6% when calculated by UV-Vis analysis. A similar preparation procedure, where BNC was soaked in *Punica granatum* peel extract and further used for nanoparticle immobilization, has been recently reported [46].

### 3.1. FTIR Analysis

The structural characteristics of pure black raspberry extract, BNC, and BNC-BR were determined by Fourier-transform infrared (FTIR) spectroscopy, as presented in Figure 1. In the spectrum of pure BNC, a characteristic C=O stretching band corresponding to the amide group I appeared at 1646 cm^−1^ and a band located at 1550 cm^−1^ was specific to the NH groups in amide II, both bands confirming the presence of residual proteins of BCN [47], while characteristic -OH stretching bands were detected at a wavenumber of 3000 to 3500 cm^−1^. The peak located in the area of 2800 to 2900 cm^−1^ was assigned to -CH_2_ stretching vibrations, but the C-O and -C-O-C stretching of glycoside bond ranging between 970–1160 cm^−1^ in cellulose were also clearly visible. Also, a vibration band at 1431 cm^−1^ was attributed to the O-C-H and H-C-H deformation in the BNC structure. The FTIR spectrum of black raspberry extract indicated the number of functional groups from different molecules such as polysaccharides, fatty acids, amino acids, phenols, etc. The intense broad band located in the range of 3000 to 3500 cm^−1^ was a typical absorption band of the -OH stretching vibration, and a characteristic -CH_2_ stretching band was confirmed at 2870 to 2930 cm^−1^. Moreover, characteristic carbonyl peaks of 1715 and 1601 cm^−1^ were attributed to the ester carbonyl (COOR) groups and carboxylate ion stretching band (COO^−^), respectively [48]. The adsorption of black raspberry extract as an active compound resulted in changes in the characteristic absorbance bands. The typical *β*-1,4 glycoside bonds’ absorption band at 893 cm^−1^ of the BNC repeating units were observed in the BNC spectra, but this characteristic band was overlapped by the BR characteristic peaks in the area 840 to 920 cm^−1^, proposing the BR molecule adsorption onto the BNC surface rich in hydroxyl groups [49]. The SO_3_^−^ symmetric stretching band appearing in the area of 750 to 800 cm^−1^ in the BR spectra was preserved in the BNC-BR [48]. In the BNC-BR FTIR spectrum, both BNC and BR characteristic peaks were preserved, but, also, the shifting of the characteristic peaks and its broadening pointed to the interactions between the BNC and BR functional groups. Hence, the appearance of the -OH stretching band (located in the range of 300–3500 cm^−1^) could be related to the formation of hydrogen bonding between BNC and BR. Also, it is possible that BNC and BR established hydrophobic interactions or Van der Waals forces [50]. The additional confirmation of established interactions between those two components was the shifting of the carbonyl characteristic peak from 1716 to 1723 cm^−1^.

### 3.2. Optical Microscopy and SEM Analysis

Micrographs captured by optical microscopy provided the preliminary insight into BNC and BNC-BR surface morphology. The compact, fibrillary, networking structure of BNC was confirmed, while black raspberry extract was perfectly adsorbed onto the BNC network (Figure 2a).

SEM inspection of the films revealed a long, fibrillary network morphology characteristic of BNC [51]. Characteristic BNC membranes with a compact, densely packed, and entangled structure, with void spaces randomly distributed throughout the matrix, were observed (Figure 2b). This highly entangled fiber network provides a large surface area and the porous structure of the BNC membranes suitable for the entrapment and immobilization of a variety of active compounds [50]. After the black raspberry adsorption into BNC, significant changes in morphology were observed. From the presented BNC-BR micrographs, the three-dimensional fibrillary network on the surface of the BNC membrane stayed intact but, in general, the morphology was changed due to the black raspberry particle aggregation on the microfibrils. In comparison to neat BNC, beans and BR particles on the surface of BNC-BR films are clearly visible. This might be because of the established intermolecular interactions between BR and BNC, which further affected the formed hydrogen bond between BNC molecules [52]. The remarkable change in the morphology of BNC-BR could also be attributed to the additional absorption of the liquid medium containing dissolved black raspberry extract by the freeze-dried procedure, which resulted in a decrease in networking, and a quite dense, flatter structure. However, the obtained structure appeared to be a potentially great material capable of preventing fluid permeability.

### 3.3. In Vitro Release of Cyanidin-3-O-Rutinoside

The release profile of cyanidin-3-O-rutinoside from lyophilized BR and BNC-BR was presented in Figure 3A, and it was evident that CR was rapidly released from both samples. The similarity factor *f*_2_ (39.37) indicated that the release profiles of CR from lyophilized BR and BNC-BR were significantly different. The slightly faster release of CR from BNC-BR than from lyophilized BR could be explained by the structure of the samples. According to the SEM analysis, BR was adsorbed on the large and porous surface of BNC membranes that has enabled the immediate contact of the active compounds and dissolution medium, but also due to the interaction between BNC and BR, confirmed by FTIR analysis. On the other hand, lyophilized BR powder contained bigger particles of uneven size, and, therefore, the CR release was slightly slower. The higher content of CR in the 0.1 M HCl than at pH 6.8 could also be attributed to the pH-dependent color and stability of anthocyanins [15]. It is known that increasing the pH leads to a decrease in the concentration of the red flavylium cation (2-phenylchromenylium cation), due to the hydration which, in turn, reduces the color intensity and produces the colorless carbinol pseudobase (hemiacetal or chromenol) [53]. Since the same HPLC method as well as wavelength (520 nm) were utilized for the determination of CR content in the samples analyzed under different pH conditions, a slightly higher CR content was detected at a lower pH (1.2 and 0.1 M HCl) than in purified water (pH ≈ 4.5), where the total CR content was determined, due to the increased anthocyanin solution color intensity at a pH between 1–3. Namely, with regard to a wide pH range (1–14), it is known that anthocyanins exist in five different structural forms in the equilibrium state (Figure 4) [54], characterized by different colors (flavylium cation as red, carbinol pseudo base as colorless, quinoidal base as purple and blue, or chalcone as yellowish), whereby the red flavylium cation is dominant in the acidic medium (pH = 1–3) [53,55]. Increasing the pH leads to a decrease of the flavylium cation concentration, followed by the reduced color intensity of the solution [55]. Actually, in low acidic or neutral to low alkaline pH medium, each anthocyanin represents a mixture of equilibrium forms, as was previously reported in our study [56]. Furthermore, the release of CR from BNC-BR was immediate at different pH values relevant to the gastrointestinal tract (Figure 3B).

According to the results presented in Figure 3B, the release of CR from BNC-BR was immediate at different pH values relevant to the gastrointestinal tract. Furthermore, CR was dissolved following the Fickian diffusion mechanism, since the release exponent (*n*) was below 0.45 at pH values from 1.2 to 7.4 (Table 2). The Korsmeyer–Peppas model was the most appropriate model for describing the CR release kinetics at different pH values, since the highest correlation coefficient (0.5220–0.6988) was reported for this model. The data on the effect of pH on the release of anthocyanins incorporated in bacterial nanocellulose are scarce, but are reported for some other natural compounds, such as α-mangostin and curcumin. Taokaew et al. [57] prepared bacterial cellulose films incorporated with the ethanolic extract of mangosteen peel. The absorption and release profiles were studied based on total phenolics and the main active compound, α-mangostin. Immersion time and concentration, as well as the pH of the dissolution medium, affected the release of bioactive compounds. In the case of α-mangostin, the phosphate buffer (pH 7.4) facilitated release more than the acetate buffer (pH 5.5). Bakr et al. [58] studied the co-delivery of fluorouracil and curcumin from nanocomposites. An acidic pH (5.4) was more supportive for the release compared to an alkaline pH (7.4) after 24 h. Moreover, it was noticed that the thickness of BNC-BR has influenced the release profile of CR (Figure 3C). The similarity factor *f*_2_ (27.95) showed that the release profile from BNC-BR 150 was significantly different from BNC-BR 250. This result has suggested that, by modifying the thickness of BNC, the release profile of the active compound could be affected. Comparably, Peres et al. [59] tested the practicability of bacterial cellulose as a drug delivery system, and noticed that the drug incorporation efficiency correlates with the membranes’ structural properties, such as the thickness, surface area, and fiber amount.

### 3.4. Total Bioactive Compound Content in Black Raspberry Pomace Extract

It is well-known that fruits, particularly berries, are one of the most important sources of phenolic compounds, such as phenolic acids, flavonoids, and tannins, that have been connected with protective effects on human health [60]. According to numerous reports, *R. occidentalis* fruit is rich in phenolics, with anthocyanins as their dominant components [22,23], appreciated for their anti-inflammatory, antioxidant, vasoprotective, as well as hypoglycemic and hypolipidemic effects [61,62]. The content of the bioactive compounds in BR determined in this study is presented in Table 3. Spectrophotometrically, the Folin–Ciocalteu assay revealed that black raspberry pomace hydro-ethanolic lyophilized extract had a valuable total phenolic content (77.25 mg GAE/g), comparable with the TPC reported for black raspberry fruit water lyophilized extract (45.15 mg GAE/g) [63]. Additionally, the significant total anthocyanins content (27.42 mg CGE/g) determined in BR is approximately 1.4-fold higher than in the 80% ethanol extract of dried black raspberry fruit from Korea [64], which can be explained by the impact of genetic, ripening-stage, environmental, and other factors [65].

### 3.5. Antioxidant Activity of Black Raspberry Extract and BNC-BR

The antioxidant potential of the black raspberry extract was evaluated for its free-radical scavenging ability (DPPH), as well as reducing power (FRAP). BR exhibited a strong and concentration-dependent DPPH radical scavenging ability (*IC*_50_ 60.51 µg/mL), along with a notable reducing capacity (FRAP value of 1.66 mmol Fe^2+^/g) (Table 2), and this was generally in agreement with literature data [63,65]. Ozgen et al. (2008) reported averaged values of 2.92 ± 0.29 in the DPPH test and 4.62 ± 0.88 mmol TE·100 g^−1^ fresh weight in the FRAP test they performed in 19 samples of four black raspberry Midwestern cultivars obtained from different production sites [65]. Shin et al. (2018) reported *IC*_50_ values from 95.9 to 824.93 µg/mL obtained in the DPPH test performed with pre-matured black raspberry which were harvested from late May to early June [63]. The antioxidant activity of the positive control, ascorbic acid, was higher that the investigated extract, according to the results of the DPPH radical scavenging ability (*IC*_50_ 4.45 ± 0.05 µg/mL) and reducing capacity (FRAP value of 15.94 mmol Fe^2+^/g). Taking into account the relationships among the antioxidant capacity and anthocyanins levels, it seems that the main contributors to the antioxidant activity of black raspberry are cyanidin derivatives (cyanidin-3-rutinoside and cyanidin-3-xylosylrutinoside) [65]. It was reported that the pure BNC membrane does not possess considerable antioxidant activity [66], which indicates that the incorporation of the BR into the BNC, explained in the present study, is a good strategy to obtain a natural material with nutritional and pharmacological features, as was previously confirmed for some other plant extracts [67,68].

### 3.6. Antimicrobial Activity Assay

The bacterial activity and the antifungal properties of the tested BR extract and BNC-BR materials were evaluated through the disk diffusion assay (Figure 5). Growth inhibition zones around the BR extract was detected for *L. monocytogenes, S. aureus,* and *E. coli* at the highest amount of the used extract (1000 µg/disc), while activity against *P. aeruginosa* and antifungal activity tested on *C. albicans* could not be indicated due to the absence of inhibition zones (Figure 5A). On the other side, BNC-BR materials showed slight inhibition zones with *L. monocytogenes* and *S. aureus* (Figure 5B), showing the efficient dynamics of BR release. BNC itself showed no antimicrobial effect (Figure 5B).

## 4. Conclusions

In this study, a two-step process for the upcycling of lignocellulose into BNC has been demonstrated. Firstly, the enzymatic treatment of organosolv-pretreated beechwood pulp resulted in a mixture of fermentable sugars that efficiently supported growth, and of *K. mendeliensis* and the subsequent production of BNC. This material was further enriched by black raspberry pomace still full of antioxidants and bioactive compounds. We further demonstrated that the new material had the ability to release antioxidants at a wide range of pH values and showed evidence of its antimicrobial properties. Overall, this process and these materials should be further examined in sustainable food production as edible materials or for functional packaging purposes.

## Data Availability

The data presented in this study are available upon request from the corresponding author.

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
