# Peer review of "Two-Step Upcycling Process of Lignocellulose into Edible Bacterial Nanocellulose with Black Raspberry Extract as an Active Ingredient"

_foods, 2023, doi:10.3390/foods12162995_

Round 1

Reviewer 1 Report

Comments and Suggestions for Authors

Comments

The manuscript titled “Upcycling lignocellulose into edible bacterial nanocellulose
with black raspberry extract as an active ingredient
” by Ponjavic et al aims the production of bacterial Nanocellulose from the lignocellulosic waste materials. Although, the manuscript has presented an interesting topic, there are some minor issues/concerns that need a careful attention. The language of the article is a kind of weak and needs a careful revision. Thus, authors have an opportunity to improve the manuscript’s readability, merits as well as its understanding for the journal readers.

Abstract

1.     The research hypothesis and its conclusions are not aligned in the abstract section.

2.     The keywords need to be different than the phrases of manuscript title.

3.     More quantitative data should be provided in the abstract.

Introduction

  1. Line 49; what is the meaning of “proess”.
  2. Lines 62 & 63; some words are overlapped.
  3. In literature, a number of species/organisms are reported to degrade and ferment lignocellulose. Authors should highlight why a particular organism (Komagataeibacter medellinensis) is more promising than other agents. When the authors narrow down the literature survey to the target bacterium, there is a gap in literature. Add a paragraph about the disadvantages or lacunae of other agents/methods used for the bioconversion of lignocellulosic biomass. Further, Authors have missed relevant studies published in recent years such as listed below, among others.

Evaluation and characterization of the cellulolytic bacterium, Bacillus pumilus SL8 isolated from the gut of oriental leafworm, Spodoptera litura: an assessment of its potential value for lignocellulose bioconversion. Environmental Technology & Innovation 27 (2022) 102459. https://doi.org/10.1016/j.eti.2022.102459  

Valorization potential of a novel bacterial strain, Bacillus altitudinis RSP75, towards Lignocellulose Bioconversion: An assessment of symbiotic bacteria from the stored grain pest, Tribolium castaneum. Microorganisms 9, 1952. https://doi.org/10.3390/microorganisms9091952

Material and methods

1.     2.1. Materials and chemicals: All the chemicals procured from a company such as citric acid, formic acid, etc can be listed together.

2.     The chemical formulae of iron (III) chloride hexahydrate and iron (II) sulfate heptahydrate needs correction.

3.     What is the unit of productivity?

4.     Line 192; correct the name of the technique, “Scanning Electron Microscopic analyses”

5.     Lines 200&201; needs a careful revision. Similarly, the statements on lines 207-210 is confusing, containing repetitions thus needs a revision.

6.     Section 2.2 needs reorganization before 2.4.

7.     Lines 2; what was the exact time interval used for the sampling?

8.     Did authors perform replicates of the experiments? The statistical analysis applied to the observed data is not mentioned in the methodology.

Results and discussion

1.     Line 323; the statement needs to be supported with some citations and justification relevant to the current study.

2.     In the FTIR analysis (figure 1), substantial changes could be seen in the band range of 700-800. Authors have a chance to elaborate their results supported by some previous reports.

3.     More discussion supported by recent references is need for the changed morphology of the BNC-BR. Possible explanations for why and how it happens need to be incorporated.

4.     In figure 2, the enlarged scans are the marked properly. Moreover, the enlarged portions provided are not clear, need redraw for more clarity and resolution

5.     Authors should compare the antibacterial activities of the BR with BNC-BR.

6.     Figure 3, can be redrawn for more clarity and higher resolution.

Conclusions

  1. The conclusion section should be rewritten in terms of your innovations or new discoveries. I wonder why authors didn’t provide the limitations as well as future directions of the research at least in the conclusion section.
Comments on the Quality of English Language

Many of the sentences are very long and confusing. Hence, authors can  revise and edit the sentences for proper meaning and better understanding. 

Reviewer 2 Report

Comments and Suggestions for Authors

Please find my comments below:

What volume of phosphate-citrate buffer was added to the 1 L flask?

There is no information for what purposes this biopolymer would be used.

Reviewer 3 Report

Comments and Suggestions for Authors

Dear Authors,

Your manuscript titled ‘Upcycling lignocellulose into edible bacterial nanocellulose with black raspberry extract as an active ingredient’ concerns a very interesting issue of the production of edible biomaterials enriched with active compounds of natural origin. However, after a thorough review of your manuscript, I have to conclude that in my opinion, it should undergo a major revision.

The general remarks that should be taken into consideration:

1.     Proofreading of the English language and style is required. The authors use long, complex sentences (even 4-5 lines), which makes it difficult for the reader to extract the proper meaning of the content (e.g., Page 2 lines 72-76; 81-85).

2.     In the Introduction section purpose and significance of the research should be more emphasized according to the current literature reports. Please insert the information that bacterial nanocellulose (BNC) can be edible and why it is digestible by humankind in comparison to cellulose itself.

3.     The section Materials and Methods should be improved and completed with indispensable procedure descriptions, formulas, and equations of standard curves

4.     The conclusion section is to short overall – after summarization of obtained results please give some details of the potential application of obtained material.

Detailed remarks that should be taken into consideration:

1.     Page 1, article title – In my opinion, the title should be revised. In the current form, it suggests that bacterial nanocellulose enriched with black raspberry extract can be directly obtained from beechwood cellulose.

2.     Page 1, abstract section – please re-write the abstract according to the ‘Instruction for Authors’ (https://www.mdpi.com/journal/foods/instructions#preparation)

‘Abstract: The abstract should be a total of about 200 words maximum. The abstract should be a single paragraph and should follow the style of structured abstracts, but without headings: 1) Background: Place the question addressed in a broad context and highlight the purpose of the study; 2) Methods: Describe briefly the main methods or treatments applied. Include any relevant preregistration numbers, and species and strains of any animals used; 3) Results: Summarize the article's main findings; and 4) Conclusion: Indicate the main conclusions or interpretations. The abstract should be an objective representation of the article: it must not contain results which are not presented and substantiated in the main text and should not exaggerate the main conclusions.’

3.     Page 1, line 33 – please insert the information about the chemical composition of cellulose and the type of glycosidic bond connecting sugar units. I suggest the implementation of a Scheme of the chemical structure of cellulose.

4.     Page 1, lines 36-37 – please insert more current data about the annual production of cellulose than from 2015 and 2017.

5.     Page 1, line 45 – I suggest changing word ‘basic’ to ‘principal’

6.     Page 2, line 54, 62 – the space between ‘production[12]’ and ‘compoundsoffering’ is needed.

7.     Page 2, line 91 – ‘The main scope ….’ – start this paragraph from a new line.

8.     Page 4, line 158, 159 – ‘ Yield (g/ml)’ and ‘…per 1L of culture medium’  - please unify if you calculate the yield on ‘ml’ or ‘L’ of culture medium’

9.     Page 4, line 162 – provide the time unit

10.  Page 4, line 173 – the numbering should be ‘2.5.1’ instead of ‘5.5.1’; I suggest to verify in the whole manuscript if you mean ‘adsorption’ or ‘absorption’ of black raspberry extract.

11.  Page 4, line 179 – EA(%) equation, please provide the explanation of particular components

12.  Page 5, line 183 – provide the equation of the standard curve and the range of concentrations used

13.  Page 5, line 194 – change ‘…nanocellulose adsorbing black raspberry …’ to ‘‘…nanocellulose with adsorbed black raspberry …’

14.  Page 5, line 220 – explanation of the components of the equation is missing, please complete

15.  Page 5, lines 221-226 – in this subsection the mathematical formulas describing these models and literature references should be given.

16.  Page 6 lines 245-249 – in addition to the literature reference, there should be a brief description of the procedure

17.  Page 7, line 271 – provide the equation of the calibration curve

18.  Page 7, line 291 – ‘1.3 mg/mL’ – I suggest changing this to ‘1.3 mg per 1mL of culture medium’

19.  Page 7, lines 291-292 – ‘When pure glucose was used as a carbon source comparable BNC productivity was achieved [36]’ – please provide the exact value achieved in [36]

20.  Page 7, lines 296-297 – ‘Similarly, BNC soaked in Punica granatum peels extract was further used for nanoparticles immobilization [37].’ – This sentence suggests that in the paper submitted by the authors, BNC enriched with black raspberry extract was also used to immobilize nanoparticles, which is not true. This sentence should be re-write to get the proper meaning or removed from the text.

21.  Page 7, line 299 – I suggest changing the caption of Scheme 1 to ‘Conceptualization of BNC-BR production using lignocellulosic hydrolysate as carbon source.’

22.  Pages 7-8, lines 301-327 – Necessarily, both the description of the FTIR spectra and Figure 1 itself should be improved. Among others, all wavelengths or the range of wavelengths described in the text should be marked in Figure 1; the range 970 – 1160 cm-1 (line 309) is not properly marked in Figure 1; lines 321-322 ‘the -OH stretching band’ – please insert the wavelength value.

23.  Page 8, line 329 – I suggest changing the Subsection title to ‘Optical and SEM analysis’ or ‘Optical and SEM imaging’

24.  Page 8, line 336 – ‘… distributed throughout the matrix (Figure 2b)’ change to ‘… distributed throughout the matrix was observed (Figure 2b)’

25.  Page 9, line 339 – adsorption or absorption?

26.  Page 10, lines 363-364 – ‘… exist in five different  structural forms in the equilibrium state’ – these forms should be provided in the manuscript

27.  Page 10, Figure 3 – Authors are asked to explain why they got the release values over 100% in Fig.3A and B? In Fig.3A please change ‘0.1M HCl’ to the pH value (I suppose pH 1.2).

28.  Page 10, lines 379-381 – ‘Korsmeyer–Peppas model was the most appropriate model for describing the CR release kinetics at different pH values.’ – Authors are asked to insert in the manuscript the explanation in what way this model was selected. Moreover, the modeling results should be included in the main text or in additional materials

29.  Page 11, lines 398-399 – Title of Table 1 – the information from what kind of material the release of cyanidin-3-rutinoside was performed.

30.  Page 12, lines 422-423 – ‘… were generally in agreement with literature data [52, 53].’ – Authors are asked to provide the exact values from given references for comparison purposes of obtained results.

31.  Page 12, lines 432-440 – In this subsection Authors describe the antibacterial activity of black raspberry extract. However, the results obtained for enriched material (BNC-BR) are missing. In my opinion, this should be completed to give clear evidence of the biofunctionality of the obtained material. In addition, the text lacks a description of the results obtained for the P. aeruginosa strain.

Comments on the Quality of English Language

Moderate editing of English language required

Round 2

Reviewer 3 Report

Comments and Suggestions for Authors

Dear Authors

thank you for addressing most of my remarks. However, there are still a few issues to be resolved before the manuscript can be accepted for further publication steps.

1.      Page 1, lines 42 – 43 – ‘The annual production of cellulose is about 1.5 trillion tons [2, 3]’ – You left old value and old references described the issue. Please provide this more actual one from 2022 you suggest in the cover letter (https://doi.org/10.1021/acssuschemeng.1c08554) and insert it into the Reference list at the end of the manuscript

2.      Page 5, line 197 – the equation of the calibration curve and the range of concentrations used are still missing. Please insert.

3.      Page 7, line 259 – please insert the equation of the calibration curve as well.

4.      Page 7, line 275, equation (5) - The authors are asked to check the correctness of this equation, whether some brackets are missing. In the current version, the unit does not match. The left of the equation is (%) and the right is (g)?

Comments on the Quality of English Language

Minor editing of English language required

Author Response

Manuscript ID: foods-2506726.R2

Title: “Two-step upcycling process of lignocellulose into edible bacterial nanocellulose with black raspberry extract as an active ingredient “

Corresponding Author: Dr Jasmina Nikodinovic-Runic

Authors: Marijana Ponjavic, Vuk Filipovic, Evangelos Topakas, Anthi Karnaouri, Jelena Zivkovic, Nemanja Krgovic, Jelena Mudric, Katarina Savikin and Jasmina Nikodinovic-Runic

Rebuttal Letter and Answers to Referee 3

We thank the Referee for additional comments and suggestions, which are addressed in the Revised manuscript. We hope that in this form, our paper is suitable for publishing. The modified sections in the Revised manuscript 2 are written in red. The detailed answers to the Referee 3 are listed below.

Reviewer 3

Dear Authors,

thank you for addressing most of my remarks. However, there are still a few issues to be resolved before the manuscript can be accepted for further publication steps.

Query 1. Page 1, lines 42 – 43 – ‘The annual production of cellulose is about 1.5 trillion tons [2, 3]’ – You left old value and old references described the issue. Please provide this more actual one from 2022 you suggest in the cover letter (https://doi.org/10.1021/acssuschemeng.1c08554) and insert it into the Reference list at the end of the manuscript).

Answer: The annual productivity of cellulose was updated with the reference from 2022.

Query 2. Page 5, line 197 – the equation of the calibration curve and the range of concentrations used are still missing. Please insert.

Answer: The missing information referring the calibration curves are implemented.

Query 3. Page 7, line 259 – please insert the equation of the calibration curve as well.

 Answer: The calibration curve data was addressed.

Query 4. Page 7, line 275, equation (5) - The authors are asked to check the correctness of this equation, whether some brackets are missing. In the current version, the unit does not match. The left of the equation is (%) and the right is (g)?

Answer: The equation was revised, multiplication with 100 in order to express total content in % was missing.
